# PP2A-Mediated GSK3β Dephosphorylation Is Required for Protocadherin-7-Dependent Regulation of Small GTPase RhoA in Osteoclasts

**DOI:** 10.3390/cells12151967

**Published:** 2023-07-29

**Authors:** Hyunsoo Kim, Noriko Takegahara, Yongwon Choi

**Affiliations:** Department of Pathology and Laboratory Medicine, University of Pennsylvania Perelman School of Medicine, Philadelphia, PA 19104, USA; hyunsoo3@pennmedicine.upenn.edu (H.K.); tnoriko@pennmedicine.upenn.edu (N.T.)

**Keywords:** osteoclasts, Pcdh7, PP2A, GSK3β, RhoA, differentiation

## Abstract

Protocadherin-7 (Pcdh7) is a member of the non-clustered protocadherin δ1 subgroup of the cadherin superfamily. Pcdh7 has been revealed to control osteoclast differentiation by regulating Rho-family small GTPases, RhoA and Rac1, through its intracellular SET binding domain. However, the mechanisms by which small GTPases are regulated downstream of Pcdh7 remain unclear. Here, we demonstrate that protein phosphatase 2A (PP2A)-mediated dephosphorylation of Glycogen synthase kinase-3β (GSK3β) is required for Pcdh7-dependent activation of RhoA during osteoclast differentiation. Pcdh7-deficient (Pcdh7^−/−^) cells showed impaired PP2A activity, despite their normal expression of PP2A. GSK3β, whose activity is regulated by its inhibitory phosphorylation at Ser9, was dephosphorylated during osteoclast differentiation in a Pcdh7-dependent manner. Inhibition of protein phosphatase by okadaic acid reduced dephosphorylation of GSK3β in Pcdh7^+/+^ cells, while activation of PP2A by DT−061 rescued impaired dephosphorylation of GSK3β in Pcdh7^−/−^ cells. Inhibition of GSK3β by AR−A014418 inhibited RANKL-induced RhoA activation and osteoclast differentiation in Pcdh7^+/+^ cells. On the other hand, DT-061 treatment rescued impaired RhoA activation and RANKL-induced osteoclast differentiation in Pcdh7^−/−^ cells. Taken together, these results demonstrate that PP2A dephosphorylates GSK3β and thereby activates it in a Pcdh7-dependent manner, which is required for activation of small GTPase RhoA and proper osteoclast differentiation.

## 1. Introduction

Osteoclasts are large cells with multiple nuclei that specialize in resorbing bone [1,2]. They originate from myeloid precursors and are of hematopoietic origin. Osteoclast differentiation is primarily triggered by the osteoclast differentiation factor, Receptor activator of nuclear factor-kB ligand (RANKL), which is mainly provided by osteocytes and osteoblasts [3,4,5,6]. However, the optimal activation and differentiation of osteoclasts require additional cell surface molecules, such as cell adhesion molecules, which mediate various multistep processes by regulating cell–cell recognition/adhesion and intracellular signal transduction [7,8,9,10].

Protocadherin-7 (Pcdh7) is a member of the protocadherins, a subgroup of calcium-dependent cell adhesion proteins, known as the cadherin superfamily [11]. Protocadherins have conserved cadherin motifs in their extracellular region but lack the intracellular conserved motifs crucial for mechanical adhesion. They appear to have varied functions as mediators of cell adhesion or regulators of other molecules [12]. Pcdh7 has been found to regulate various physiological and pathological processes not only by controlling cell adhesion but also by regulating cell signaling [13,14,15,16,17,18,19]. Recently, we demonstrated the requirement of Pcdh7 in osteoclast differentiation. Pcdh7 plays a crucial role in the activation of Rho-family small GTPases, RhoA and Rac1 in osteoclasts. Its intracellular domain, which is known for binding to the oncogene SET (SET/Template-activating factor 1 (TAF1)), is essential for regulating RhoA and Rac1 [20,21]. These findings provide evidence that Pcdh7 plays a role in osteoclast differentiation as a mediator of cell signaling. However, it remains unclear how Pcdh7 controls the activation of RhoA and Rac1 through its intracellular SET binding domain during osteoclast differentiation.

Pcdh7 has been reported to functionally interact with intracellular signaling molecules, not only SET but also protein phosphatase 1alpha (PP1α) and protein phosphatase 2A (PP2A) [22,23,24]. The interaction between PP2A and SET has been reported in cell lines [23,25]. The activity of PP2A and Rho-family small GTPase is reportedly linked through Glycogen synthase kinase-3β (GSK3β) in cancer cells [26,27,28]. Since Pcdh7 associates with RhoA and Rac1 through its intracellular SET binding domain and since the domain is critical for Pcdh7-mediated activation of RhoA and Rac1 during osteoclast differentiation [20], these observations suggest that PP2A plays a relevant role in Pcdh7-mediated signaling in osteoclasts.

In this study, we identify PP2A as a signaling molecule that mediates Pcdh7-dependent regulation of RhoA and osteoclast differentiation by regulating GSK3β activation. Pcdh7 deficiency exhibited impaired PP2A activity in addition to defective dephosphorylation of GSK3β during osteoclast differentiation. Inhibition of GSK3β abolishes RANKL-induced RhoA activation and osteoclast differentiation in Pcdh7^+/+^ cells. On the other hand, activation of PP2A rescues defective GSK3β dephosphorylation, RhoA activation, and osteoclast differentiation in Pcdh7^−/−^ cells. Taken together, our findings reveal that the Pcdh7–PP2A–GSK3β signaling axis regulates RhoA during osteoclast differentiation.

## 2. Materials and Methods

### 2.1. In Vitro Cell Culture

To prepare BMMs and osteoclasts, we followed the previously described protocol [20] using wild-type (Pcdh7^+/+^) and Pcdh7-deficient (Pcdh7^−/−^) mice. The femurs and tibias were extracted, and the whole bone marrow was incubated in α−MEM (Invitrogen, Carlsbad, CA, USA) medium containing 10% fetal bovine serum and M−CSF (5 ng/mL) overnight in 100 mm petri dishes. Non-adherent cells were collected, and BMMs were generated by culturing the cells with M−CSF (60 ng/mL) for three days. To prepare preosteoclasts, BMMs were cultured with M−CSF (60 ng/mL) and RANKL (150 ng/mL) for one day. For osteoclast differentiation, BMMs were cultured with M−CSF (60 ng/mL) and RANKL (150 ng/mL) for three days. Osteoclasts were stained using Leukocyte acid phosphatase Lit (387A−1KT, Sigma, St. Louis, MO, USA) according to the manufacturer’s instructions. Okadaic acid (Cat.# ab120375) was purchased from abcam (Cambridge, MA, USA); Cytostatin (Cat.# 19602), LB−100 (Cat.# 29105), and Rubratoxin A (Cat.#19605) were purchased from Cayman chemical (Ann Arbor, MI, USA); AR−A014418 (Cat.# A3230) was purchased from Sigma (St. Louis, MO, USA); and DT−061 (Cat.# S8774) was purchased from Shelleckchem, Houston, TX, USA.

### 2.2. Pull-Down Assay and Western Blotting

The active form of RhoA was detected using the RhoA/Rac1/Cdc42 Activation Assay Combo Biochem kit (Cat.# BK030, Cytoskeleton Inc., Denver, CO, USA), following the manufacturer’s instructions. For western blotting, cell cultures were lysed with lysis buffer containing a protease and phosphatase inhibitor cocktail, and protein concentrations were determined using the Bradford assay. The proteins were then subjected to electrophoretic resolution and transferred to PVDF membranes. Western blotting was performed with the following antibodies: anti-PP2A−A, anti-PP2A−B, and anti-PP2A−C [Cat.# 9780T, Cell Signaling Technology (CST), Beverly, MA, USA], anti-Pcdh7 (Cat.# TA505452, Origene, Rockville, MD, USA), anti-phopho-GSK3β (Ser9) [Cat.# 9336, Cell Signaling Technology (CST), Beverly, MA, USA], anti-GSK3β [Cat.#, 9315; Cell Signaling Technology (CST), Beverly, MA, USA], and anti-actin (Cat.# sc-47778, Santacruz Biotechnology; Santa Cruz, CA, USA).

### 2.3. Measurement of PP2A Activity

PP2A activity was measured by using the Ser/Thr Phosphatase assay kit (Cat. # 17-127, Millipore, Billerica, MA, USA) according to the manufacturer’s instructions.

### 2.4. Statistical Analysis

A one-way Anova or 2-tailed paired Student’s *t* test were used to determine the significance of differences by Prism 9.5.1 (GraphPad Software Inc., San Diego, CA, USA). *p* < 0.05 was considered statistically significant.

## 3. Results

### 3.1. Pcdh7 Deficiency Results in Impaired PP2A Activity in Osteoclasts

First, we sought to identify the involvement of PP2A in the Pcdh7-dependent regulation of osteoclast differentiation. PP2A is a multi-subunit enzyme composed of three subunits termed catalytic (C), scaffolding (A), and regulatory (B) subunits [29]. We examined the protein expression levels of PP2A enzyme subunits in Pcdh7^+/+^ and Pcdh7^−/−^ cells. Mouse-bone-marrow-derived monocytes (BMMs) from Pcdh7^+/+^ and Pcdh7^−/−^ mice were treated with M−CSF + RANKL for one day to induce preosteoclasts, and the expression levels of PP2A subunits were determined by western blotting. The protein expression levels of these subunits were comparable between BMMs and preosteoclasts, and their expression was not affected by the absence of Pcdh7 (Figure 1A). We then examined PP2A activity in Pcdh7^+/+^ and Pcdh7^−/−^ cells. Whole-cell lysates from Pcdh7^+/+^ and Pcdh7^−/−^ cells were prepared, and PP2A activity was measured. Comparable levels of PP2A activity were observed between Pcdh7^+/+^ and Pcdh7^−/−^ BMMs (Figure 1B). PP2A activity was enhanced by RANKL stimulation in Pcdh7^+/+^ cells that peaked on Day 1, but this enhancement was significantly diminished in Pcdh7^−/−^ cells (Figure 1B). These results suggested that PP2A activity is downregulated in the absence of Pcdh7 during osteoclast differentiation. To understand the role of PP2A during osteoclast differentiation, we cultured wild-type BMMs with M−CSF + RANKL for three days to induce osteoclasts in the presence of a protein phosphatase inhibitor okadaic acid that has a high affinity for PP2A [30]. We found that RANKL-induced osteoclast differentiation was significantly inhibited by okadaic acid in a dose-dependent manner (Figure 1C). We also examined the effect of additional PP2A-specific inhibitors, namely Cytostatin, LB−100, and Rubratoxin A, and found that these inhibitors significantly inhibited osteoclast differentiation in a dose-dependent manner (Appendix A). These results suggest an important role of PP2A in the Pcdh7-mediated regulation of osteoclast differentiation.

### 3.2. Pcdh7 Deficiency Results in Impaired PP2A−Mediated GSK3β Activation in Osteoclasts

PP2A has been shown to regulate GSK3β activity. The activity of GSK3β is regulated by inhibitory phosphorylation at Ser9, and PP2A dephosphorylates it [26,31,32]. When wild-type BMMs were cultured with M−CSF + RANKL for three days in the presence of GSK3β-specific inhibitor AR−A014418, osteoclast differentiation was significantly inhibited in an AR−A014418 dose-dependent manner (Figure 2A), suggesting a role of GSK3β in osteoclast differentiation. We thus sought to identify whether GSK3β is dephosphorylated by PP2A in a Pcdh7-dependent manner. We first examined phosphorylation of GSK3β in the presence or absence of Pcdh7. Pcdh7^+/+^ and Pcdh7^−/−^ BMMs were treated with M−CSF and RANKL for three days to induce osteoclasts, and phosphorylation of GSK3β was determined by western blotting. Dephosphorylation of GSK3β was observed during differentiation and peaked on day 1 after RANKL treatment (Figure 2B). On the other hand, such dephosphorylation of GSK3β was diminished in the absence of Pcdh7 (Figure 2B). These results suggested a requirement of Pcdh7 for dephosphorylation of GSK3β. We next examined the role of PP2A in dephosphorylation of GSK3β by using the inhibitor or activator of PP2A. When Pcdh7^+/+^ preosteoclasts were treated with okadaic acid, dephosphorylation of GSK3β was inhibited (Figure 2C). On the other hand, when Pcdh7^−/−^ preosteoclasts were treated with PP2A-specific activator DT-061, dephosphorylation of GSK3β was enhanced to the same degrees as Pcdh7^+/+^ preosteoclasts (Figure 2D). These results suggest that GSK3β is dephosphorylated at Ser9 in a Pcdh7-dependent manner, and the dephosphorylation is mediated by PP2A.

### 3.3. Pcdh7-Dependent RhoA Activation Requires PP2A and GSK3β

We sought to further determine the role of PP2A-mediated dephosphorylation of GSK3β in Pcdh7-dependent osteoclast differentiation. GSK3β has been shown to regulate Rho-family small GTPases [28]. We examined whether PP2A−GSK3β regulates the activation of RhoA in osteoclasts. Pcdh7^+/+^ preosteoclasts were stimulated with M−CSF + RANKL in the presence or absence of a GSK3β inhibitor, and the activation of RhoA was determined by a pull-down assay. As expected, RANKL stimulation induced activation of RhoA (Figure 3A) [20], while the activation of RhoA was abolished by treatment with the GSK3β inhibitor (Figure 3A). These results suggested a requirement for GSK3β activation in RANKL-induced RhoA activation. Next, Pcdh7^−/−^ preosteoclasts were stimulated with M−CSF + RANKL in the presence or absence of PP2A activator, and RhoA activation was examined. RANKL-induced RhoA activation was observed in Pcdh7^+/+^ cells, while it was impaired in Pcdh7^−/−^ cells (Figure 3B), consistent with the previous study [20]. Notably, the impaired RhoA activation in Pcdh7^−/−^ cells was rescued when treated with the PP2A activator (Figure 3B). Additionally, treatment with the PP2A activator rescued impaired osteoclast differentiation in Pcdh7^−/−^ cultures (Figure 3C). Collectively, these results suggest that PP2A−GSK3β activation is required for the Pcdh7-dependent activation of the small GTPase RhoA, which is critical for osteoclast differentiation.

## 4. Discussion

In this study, we aimed to fill a gap in previous research by investigating how Pcdh7 regulates the activation of small GTPase RhoA through its intracellular SET binding domain. We demonstrated that PP2A-mediated activation of GSK3β by dephosphorylation of its Ser9 is required for the Pcdh7-dependent regulation of RhoA activation during osteoclast differentiation (Figure 4). We revealed that Pcdh7 deficiency exhibited a significant reduction in PP2A activity in preosteoclasts but not in BMMs, although the protein expression levels of PP2A subunits were comparable among Pcdh7^+/+^ and Pcdh7^−/−^ BMMs and preosteoclasts. These results suggested that the expression of these subunits is unaffected by RANKL stimulation and does not depend on the presence of Pcdh7. Previously, we showed that the expression of Pcdh7 protein was induced by RANKL stimulation and peaked in preosteoclasts cultured with RANKL for one day [21]. Since PP2A activity reached its peak on day 1 following RANKL stimulation, these findings suggest a specific requirement for Pcdh7 for PP2A activation during osteoclast differentiation. In addition, we revealed its requirement for GSK3β dephosphorylation during osteoclast differentiation. Dephosphorylation of GSK3β peaked on day 1 after RANKL stimulation, consistent with the expression profiles of Pcdh7 and PP2A activity. Impaired dephosphorylation of Pcdh7^−/−^ cells could be rescued by the PP2A-specific activator DT−061. Additionally, DT-061 treatment recovered impaired RhoA activation and osteoclast differentiation in Pcdh7^−/−^ cells. Our findings provide evidence of the requirement of PP2A for the Pcdh7-dependent regulation of RhoA during osteoclast differentiation. Although this study only showed the activation of RhoA, not Rac1, it is plausible that Pcdh7-dependent Rac1 activation is also regulated through the PP2A−GSK3β pathway, given that previous research has demonstrated that Pcdh7-dependent activation of RhoA and Rac1 exhibits a similar behavior [20].

Previous studies have demonstrated the association between Pcdh7 and SET, SET and PP2A, and Pcdh7 and PP2A [23,25], suggesting that these molecules form a complex together. Pcdh7 has also been reported as interacting with PP1α [22,24]. We revealed a critical role of PP2A in the Pcdh7-dependent regulation of osteoclast differentiation. However, the involvement of PP1α in Pcdh7-mediated osteoclast regulation remains unclear. Although okadaic acid has the capacity to inhibit PP1α, the complete inhibition of PP1α requires a much higher concentration of okadaic acid than that required for PP2A [33]. This suggests that the inhibitory effect of okadaic acid is mostly due to the inhibition of PP2A and not PP1α. Additionally, we showed that three additional PP2A-specific inhibitors inhibited osteoclast differentiation to the same extent as okadaic acid (Appendix A). Furthermore, we utilized the PP2A-specific activator DT–061 and demonstrated that it restored impaired RhoA activation and osteoclast differentiation in Pcdh7^−/−^ cells. These findings suggest that PP2A assumes the role of the Pcdh7-mediated regulation of osteoclast differentiation. Nevertheless, we cannot exclude the possibility of involvement of other protein phosphatases including PP1α in the Pcdh7-mediated regulation of osteoclast differentiation. Future studies will be required to reveal the contribution of PP1α and other protein phosphatases in the Pcdh7-mediated regulation of osteoclast differentiation.

SET is known to be a potent physiological inhibitor of PP2A [34,35]. In transformed cell lines, it has been demonstrated that SET forms a protein complex with Pcdh7 and PP2A, which leads to the inhibition of PP2A activity. The suppression of PP2A activity results in an enhanced activation of ERK, which is considered to be a contribution of Pcdh7 to tumorigenesis [23]. In contrast, in osteoclasts, we revealed that Pcdh7 deficiency resulted in a significant reduction of PP2A activity. We examined the phosphorylation of ERK in BMMs and osteoclasts in the presence or absence of Pcdh7 and found no significant change in the phosphorylation of ERK in Pcdh7^−/−^ cells (Appendix A). These results suggested that the Pcdh7-dependent inhibition of PP2A, followed by the enhanced activation of ERK, is unlikely to occur in osteoclasts. It remains unclear how Pcdh7 activates PP2A in osteoclasts. It is possible that cell-type-specific mechanisms regulate PP2A activity downstream of Pcdh7. Future investigations are necessary to address this issue.

In this study, we demonstrated the requirement of the activation of GSK3β for Pcdh7-dependent osteoclast differentiation, as evidenced by (1) the impaired dephosphorylation of GSK3β during osteoclast differentiation in Pcdh7-deficient cells, and (2) the fact that the GSK3β inhibitor abolished RANKL-induced RhoA activation, which is required for Pcdh7-dependent osteoclast differentiation [20]. It has been reported that the inactivation of GSK3β, which is mediated by the PI3K–AKT signaling pathway, induces NFATc1 expression and regulates osteoclast differentiation [36,37]. We examined the activation of AKT and expression of NFATc1 during osteoclast differentiation in the presence or absence of Pcdh7 and found comparable levels of activation of AKT and expression of NFATc1 between Pcdh7^+/+^ and Pcdh7^−/−^ cells (Appendix A). These findings suggested that impaired osteoclast differentiation in Pcdh7 deficiency is unlikely to be attributed to the inactivation of AKT signaling and NFATc1 expression.

Taken together, we identified the Pcdh7–PP2A–GSK3β signaling axis in regulation of RhoA activation, which contributes to RANKL-induced osteoclast differentiation.

## 5. Conclusions

Multinucleation is a hallmark of mature osteoclasts. Pcdh7 deficiency resulted in the impaired formation of multinucleated osteoclasts. We have previously demonstrated that Pcdh7 plays a crucial role in the activation of RhoA in osteoclasts, and its intracellular SET binding domain is essential for regulating RhoA activity. In this study, we aimed to address a gap in previous research, specifically focusing on how Pcdh7 controls the activation of RhoA through its intracellular SET binding domain. Our findings are as follows: (1) We identified the requirement of Pcdh7 for the optimal activation of PP2A. (2) We established the necessity of PP2A activation for GSK3β activation through dephosphorylation at Ser9. (3) We highlighted the significance of GSK3β activation for RhoA activation in osteoclasts. These results demonstrate that PP2A dephosphorylates GSK3β, thereby activating it in a Pcdh7-dependent manner, which is essential for the activation of the small GTPase RhoA and proper osteoclast differentiation.

## Figures and Tables

**Figure 1 cells-12-01967-f001:**
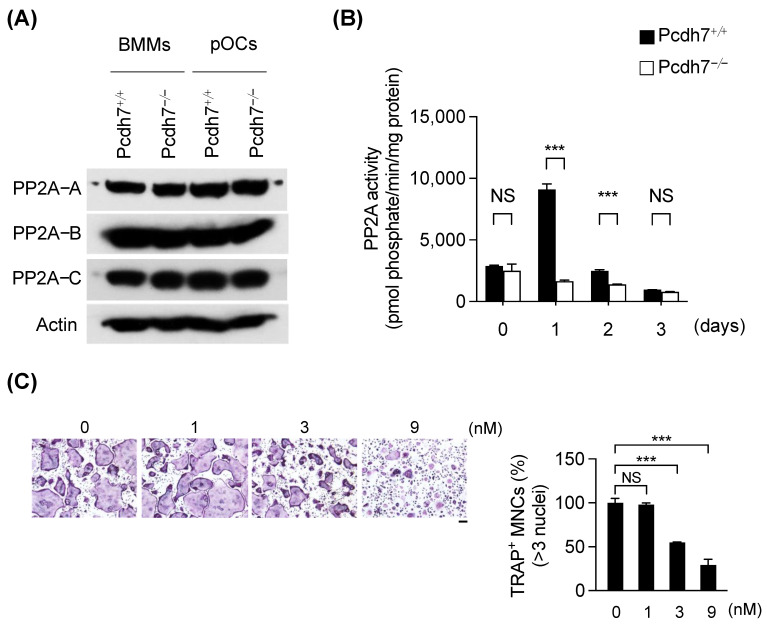
Pcdh7 deficiency results in impaired PP2A activity in osteoclasts. (**A**) Expression levels of PP2A subunits in Pcdh7^+/+^ and Pcdh7^−/−^ BMMs and preosteoclasts. Cell lysates were analyzed by western blotting using indicated antibodies. (**B**) PP2A activity in Pcdh7^+/+^ and Pcdh7^−/−^ cells. Horizontal axis indicates days after RANKL stimulation. Day 0 indicates BMMs. PP2A enzyme activity was determined in whole-cell lysates. (**C**) Effect of protein phosphatase inhibitor okadaic acid on osteoclast differentiation. Pcdh7^+/+^ BMMs were treated with M−CSF + RANKL for three days in the presence of the indicated amount of okadaic acid. The percentage of TRAP^+^ multinucleated cells (three nuclei or more per cell) is shown (right). The scale bar represents 100 μm. Data are shown as the mean ± S.D. *** *p* < 0.001. NS—not significant. Data are representative of three experiments.

**Figure 2 cells-12-01967-f002:**
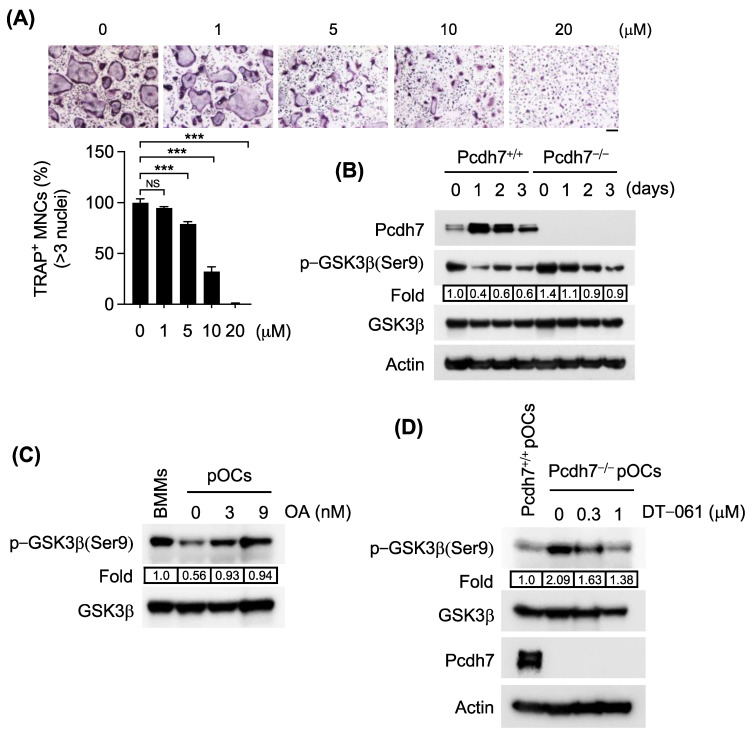
Pcdh7 deficiency results in impaired PP2A-mediated GSK3β activation in osteoclasts. (**A**) Effect of the GSK3β inhibitor AR−A014418 on osteoclast differentiation. Pcdh7^+/+^ BMMs were treated with M−CSF + RANKL for three days in the presence of the indicated amount of AR−A014418. The percentage of TRAP^+^ multinucleated cells (three nuclei or more per cell) is shown (bottom). The scale bar represents 100 μm. Data are shown as the mean ± S.D. *** *p* < 0.001. NS—not significant. (**B**) Phosphorylation of GSK3β in the presence or absence of Pcdh7. BMMs from Pcdh7^+/+^ and Pcdh7^−/−^ mice were cultured with M−CSF + RANKL for up to three days, and western blotting was performed with the indicated antibodies. (**C**) Pcdh7^+/+^ preosteoclasts were treated with the indicated amount of okadaic acid, and phosphorylation of GSK3β was detected by western blotting. BMMs were used as a control. (**D**) Pcdh7^−/−^ preosteoclasts were treated with the indicated amount of PP2A activator DT−061, and phosphorylation of GSK3β was detected by western blotting. Pcdh7^+/+^ preosteoclasts were used as a control. Data are representative of three experiments.

**Figure 3 cells-12-01967-f003:**
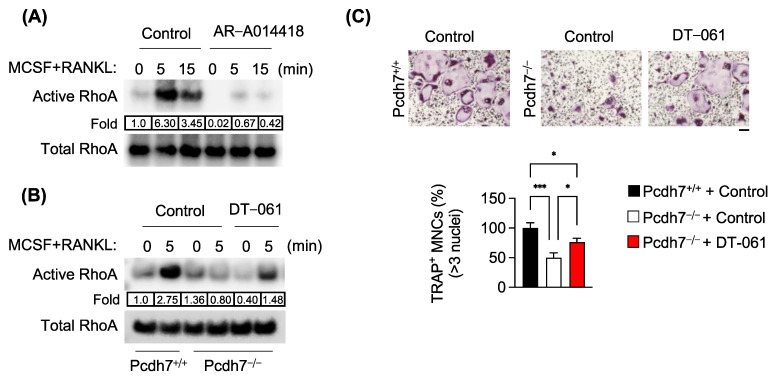
Pcdh7-dependent RhoA activation requires PP2A and GSK3β. (**A**,**B**) Effect of the GSK3β inhibitor AR−A014418 and the PP2A activator DT−061 on activation of RhoA. Pcdh7^+/+^ preosteoclasts were stimulated with M−CSF + RANKL for the indicated times in the presence or absence of (**A**) AR−A014418 (10 μM) and (**B**) DT−061 (10 μM). Whole-cell lysates were used to detect the activated form of RhoA. (**C**) Effect of DT−061 on osteoclast differentiation. Pcdh7^+/+^ and Pcdh7^−/−^ BMMs were treated with M−CSF + RANKL for three days in the presence of DT−061 (10 μM). The percentage of TRAP^+^ multinucleated cells (three nuclei or more per cell) is shown (bottom). The scale bar represents 100 μm. Data are shown as the mean ± S.D. * *p* < 0.05, *** *p* < 0.001. Data are representative of three experiments.

**Figure 4 cells-12-01967-f004:**
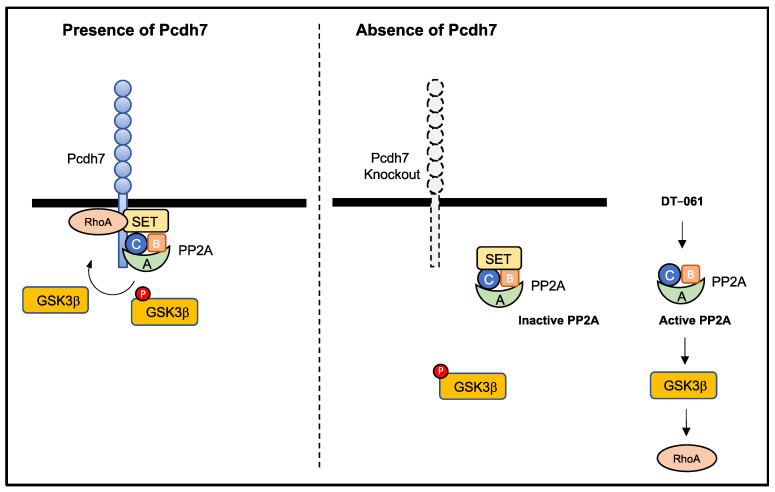
Possible model of Pcdh7–PP2A–GSK3β signaling axis in osteoclasts. Pcdh7 associates with PP2A through its association with SET. PP2A dephosphorylates GSK3β, which leads to the activation of GSK3β. Activated GSK3β induces activation of small GTPase RhoA, which subsequently promotes osteoclast differentiation. In the absence of Pcdh7, the activation of PP2A and subsequent activation of GSK3β is impaired. Treatment with the PP2A activator DT–061 can induce the activation of PP2A in the absence of Pcdh7 and rescue the impaired osteoclast differentiation in Pcdh7^−/−^ cells. A, a scaffolding subunit; B, a regulatory subunit; C, a catalytic subunit; P, phosphorylation.

## Data Availability

Not applicable.

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
