# Peer review of "PP2A-Mediated GSK3β Dephosphorylation Is Required for Protocadherin-7-Dependent Regulation of Small GTPase RhoA in Osteoclasts"

_cells, 2023, doi:10.3390/cells12151967_

Round 1

Reviewer 1 Report

In this manuscript, Kim et al. provide further insights into their previous research and indicate the regulatory role of Pcdh7 in osteoclasts by modulating RhoA via the PP2A-GSK3β axis. Several places in the manuscript should be modified or clarified for acceptance.

Fig 1B, to complement the observed induction of Pcdh7 protein expression and the decrease in GSK3β phosphorylation during osteoclast differentiation, the authors should show the dynamic changes in PP2A activity from day 0 to day 3.

A literature search has revealed that okadaic acid exhibits a significantly higher affinity for PP2A (IC50=0.1-0.3 nM) and inhibits PP1 (IC50=15-50 nM), PP3 (IC50=3.7-4 nM), PP4 (IC50=0.1 nM), and PP5 (IC50=3.5 nM). Therefore, to enhance the credibility of their conclusions, I suggest that the authors incorporate at least two different inhibitors in their experiments. Additionally, whether Pcdh7 affects the activity of other protein phosphatases?

The authors should include a discussion on the mechanism by which Pcdh7 activates PP2A.

Line 118, the authors should refer to the compound as DT-061 instead of DT-016.

In Fig 4, it is noteworthy that the authors did not investigate whether Pcdh7 recruits PP2A through SET in osteoclasts. Furthermore, previous findings have indicated that Pcdh7 associates with RhoA and Rac1 to regulate their activation. Therefore, the authors should clarify whether they consider the mechanism proposed in this study to be a correction of the previous findings or if both mechanisms exist simultaneously. It is recommended that the authors revise Fig 4 to enhance its rigor.

Author Response

Reviewer #1

The reviewer #1 mentioned that several places in the manuscript should be modified or clarified. According to the reviewer’s suggestion, we carried out suggested experiments and added explanation/discussion to address the issues, as follows.

Fig 1B, to complement the observed induction of Pcdh7 protein expression and the decrease in GSK3β phosphorylation during osteoclast differentiation, the authors should show the dynamic changes in PP2A activity from day 0 to day 3.

            As suggested by the reviewer, we examined dynamic changes in PP2A activity in Pcdh7+/+ and Pcdh7-/- cells during RANKL-induced osteoclast differentiation. We found significant increase of PP2A activity by RANKL stimulation in Pcdh7+/+ cells, that peaked on Day 1 (Revised Figure 1B). In contrast, RANKL-induced enhancement of PP2A activity was significantly diminished in Pcdh7-/- cells (Revised Figure 1B). We replaced original Figure 1B with the revised experimental results (Revised Figure 1B), and carefully described these results in the Result, Figure legends, and Discussion section (Line 83-85, Line 99-101, Line 179-181, and Line 184).

A literature search has revealed that okadaic acid exhibits a significantly higher affinity for PP2A (IC50=0.1-0.3 nM) and inhibits PP1 (IC50=15-50 nM), PP3 (IC50=3.7-4 nM), PP4 (IC50=0.1 nM), and PP5 (IC50=3.5 nM). Therefore, to enhance the credibility of their conclusions, I suggest that the authors incorporate at least two different inhibitors in their experiments. Additionally, whether Pcdh7 affects the activity of other protein phosphatases?

            As suggested by the reviewer, we prepared additional PP2A-specific inhibitors, namely Cytostatin, LB-100, and Rubratoxin A (Line 255-257), and examined the effect of these inhibitors on RANKL-induced osteoclast differentiation. We found that osteoclast differentiation was significantly inhibited by these inhibitors in a dose-dependent manner (Supplementary Figure 1). These results are consistent with the results observed in okadaic acid treatment (Figure 1C), supporting our hypothesis that PP2A assumes an important role in osteoclast differentiation. Nevertheless, we cannot exclude the possibility of involvement of other protein phosphatases including PP1a in Pcdh7-mediated regulation of osteoclast differentiation. We added these results and carefully described this issue in the Results and Discussion section (Line 91-94, Line 209-210, and Line 214-217).

The authors should include a discussion on the mechanism by which Pcdh7 activates PP2A.

             It remains unclear how Pcdh7 activates PP2A in osteoclasts. The mechanism by which Pcdh7 regulates PP2A activation could be cell type-specific, as discussed in the Discussion section (Line 218-227). Future investigation is necessary to address this issue. We carefully described about this issue in the Discussion section (Line 227-229).

Line 118, the authors should refer to the compound as DT-061 instead of DT-016.

            As suggested by the reviewer, we corrected the typo (Line 123).

In Fig 4, it is noteworthy that the authors did not investigate whether Pcdh7 recruits PP2A through SET in osteoclasts. Furthermore, previous findings have indicated that Pcdh7 associates with RhoA and Rac1 to regulate their activation. Therefore, the authors should clarify whether they consider the mechanism proposed in this study to be a correction of the previous findings or if both mechanisms exist simultaneously. It is recommended that the authors revise Fig 4 to enhance its rigor.

            We have previously demonstrated that Pcdh7 plays a crucial role in the activation of RhoA in osteoclasts, and its intracellular SET binding domain is essential for regulation of RhoA activity. In this study, we addressed a gap in the previous research, specifically how Pcdh7 controls the activation of RhoA through its intracellular SET binding domain, as described in the Introduction section (Line 51-52). We revealed: (1) the requirement of Pcdh7 for optimal activation of PP2A, (2) the necessity of PP2A activation for GSK3b activation through dephosphorylation at Ser9, and (3) the significance of GSK3b activation for RhoA activation in osteoclasts. Given the demonstrated association between SET and PP2A (Leukemia (2014) 28:1915–1918), as well as the association between Pcdh7 and PP2A, and Pcdh7 and SET (Cancer Res (2017) 77:187-197), it is plausible that Pcdh7 forms a protein complex with SET and PP2A in osteoclasts. In this study, we showed the activation of RhoA, not Rac1. However, as discussed in the Discussion section (Line 188-192), it is possible that Pcdh7-dependent Rac1 activation is also regulated through the PP2A-GSK3b pathway, given that previous research has demonstrated similar behavior in Pcdh7-dependent activation of RhoA and Rac1 (Int. J. Mol. Sci. (2021) 22: 13117). We carefully described this issue in the Discussion section (Line 169-171, Line 201-203) and revised Figure 4 to enhance comprehension.

Reviewer 2 Report

In this study, the authors found that PP2A dephosphorylates GSK3β and thereby activates it in a Pcdh7-dependent manner, which is required for activation of small GTPase RhoA and proper osteoclast differentiation. However, I do not agree its publication in " cells " due to following issues:

1. In this article, the author observed the results of osteoclast differentiation merely with the percentage of TRAP+ multinucleated cells, which is insufficient and not convincing. I suggest that you should add more experimental evidence to illustrate the changes of osteoclast differentiation.

2. In “Results” part, the author described that “The protein expression levels of these subunits were comparable between BMMs and preosteoclasts, and their expression was not affected by the absence of Pcdh7”, the author should explain the reasons in the part of “Discussion”.

3. In “2.1. Pcdh7 deficiency results in impaired PP2A activity in osteoclasts” part, the author described that “PP2A activity was enhanced in Pcdh7+/+ preosteoclasts”, Please specify to which group the enhancement is compared?

Minor editing of English language required

Author Response

Reviewer #2

The reviewer #2 mentioned that he/she does not agree this manuscript publication in Cells. We respectfully provide a rebuttal statement addressing the comments made by the reviewer.

  1. In this article, the author observed the results of osteoclast differentiation merely with the percentage of TRAP+ multinucleated cells, which is insufficient and not convincing. I suggest that you should add more experimental evidence to illustrate the changes of osteoclast differentiation.

            By generating Pcdh7-/- mice, we previously demonstrated that deficiency in Pcdh7 resulted in increased bone mass in trabecular bone (BMB Rep. (2020) 53:472-477). This effect was attributed to a reduced number of osteoclasts in vivo, while the differentiation and function of osteoblasts remained normal (BMB Rep. (2020) 53:472-477). Additionally, through an in vitro co-culture system using bone marrow stromal cells and bone marrow monocytes, we uncovered the cell-intrinsic role of Pcdh7 in formation of multinuclear osteoclasts (BMB Rep. (2020) 53:472-477). Furthermore, we revealed that the signaling pathway through Pcdh7 intracellular SET binding domain is essential for the activation of small GTPases RhoA and Rac1, which have critical roles in osteoclast differentiation (Int. J. Mol. Sci. (2021) 22: 13117). Consistent with our findings, another research group reported the involvement of Pcdh7 in osteoclastogenesis using an RNAi-mediated Pcdh7 knockdown system and assessing the presence of TRAP+ multinucleated cells. In their report, they also revealed that Pcdh7 is required for the expression of osteoclast fusion-related molecules, including Dcstamp, Ocstamp, and Atp6V0d2 (Biochem Biophys Res Commun (2014) 455:305–311). Taken together, these preceding studies strongly indicate the crucial role of Pcdh7 in the process of osteoclast differentiation.

  1. In “Results” part, the author described that “The protein expression levels of these subunits were comparable between BMMs and preosteoclasts, and their expression was not affected by the absence of Pcdh7”, the author should explain the reasons in the part of “Discussion”.

            Based on the results in Figure 1A, we can infer that the expression of these subunits is unaffected by RANKL stimulation and does not depend on the presence of Pcdh7. We have included this interpretation into the Discussion section (Line 174-177).

  1. In “2.1. Pcdh7 deficiency results in impaired PP2A activity in osteoclasts” part, the author described that “PP2A activity was enhanced in Pcdh7+/+ preosteoclasts”, Please specify to which group the enhancement is compared?

            PP2A activity was enhanced in Pcdh7+/+ preosteoclasts (stimulated with RANKL for one day) compared to Pcdh7+/+ BMMs, but this enhancement was significantly diminished in Pcdh7-/- cells. We carefully rewrote this sentence in line 83-85, and revised Figure 1B.

Reviewer 3 Report

The manuscript entitled, "PP2A-mediated GSK3β dephosphorylation is required for Protocadherin-7-dependent regulation of small GTPase RhoA in osteoclasts”, by Kim et al., has presented fascinating mechanistic pathway for osteoclast differentiation of bone marrow derived monocytes directing towards the possible therapeutics to inhibit the bone resorption. The authors have clearly shown how Pcdh7 modulates the intracellular signaling in osteoclasts formation through PP2A activity and GSK3β dephosphorylation. While, there are minor things that should be addressed by the authors:

   1. In the introduction section, the authors have mentioned that Pcdh7 functionally interact with SET as well as PP1a and PP2A. It is unclear if that interaction is physical through SET only or both SET and PP2A in a complex? It would be interesting to know the nature of interaction to form a complex. Additionally, it would clear the confusion for the reader in the discussion section (Figure 4).

2.     It would be better if the authors could provide the sample size for each experiment and quantified graph for the western blot images.

Author Response

Reviewer #3

The reviewer #3 mentioned that this study has clearly shown how Pcdh7 modulates the intracellular signaling in osteoclast formation through PP2A activity and GSK3b dephosphorylation. However, he/she raised several minor issues to be addressed. According to the reviewer’s suggestions, we added explanation/discussion to address the issues, as follows.

  1. In the introduction section, the authors have mentioned that Pcdh7 functionally interact with SET as well as PP1a and PP2A. It is unclear if that interaction is physical through SET only or both SET and PP2A in a complex? It would be interesting to know the nature of interaction to form a complex. Additionally, it would clear the confusion for the reader in the discussion section (Figure 4).

            Previous studies have demonstrated the association between Pcdh7 and SET, SET and PP2A, and Pcdh7 and PP2A (Leukemia (2014) 28:1915–1918, Cancer Res (2017) 77:187-197), suggesting that these molecules form a complex together. We carefully described this issue in the Discussion section (Line 201-203) and revised Figure 4 to enhance comprehension.

  1. It would be better if the authors could provide the sample size for each experiment and quantified graph for the western blot images.

            As suggested by the reviewer, we added the sample size for each experiment (Line 105, 139, and 166-167), and quantified western blot images (Figure 2B, 2C, 2D, 3A and 3B).

Round 2

Reviewer 2 Report

 In this article, the authors investigated the mechanisms by which small GTPases are regulated downstream of Pcdh7.  But I think that merely observing the percentage of TRAP+ multinucleated cells did not provide sufficient evidence to support the results of osteoclast differentiation in this study.  The author's response to this comment, however, was neither specific nor positive.  Were the authors considering additional tests such as F-actin staining, bone resorption pits, or expression of the calcitonin receptor, cathepsin-k, integrins αvβ3?  So, I suggest the rejection of the current manuscript.
